# Aberrant Multimodal Connectivity Pattern Involved in Default Mode Network and Limbic Network in Amyotrophic Lateral Sclerosis

**DOI:** 10.3390/brainsci13050803

**Published:** 2023-05-15

**Authors:** Haifeng Chen, Zheqi Hu, Zhihong Ke, Yun Xu, Feng Bai, Zhuo Liu

**Affiliations:** 1Department of Neurology, Nanjing Drum Tower Hospital, The Affiliated Hospital of Nanjing University Medical School, Nanjing 210008, China; 2Nanjing Drum Tower Hospital Clinical College of Traditional Chinese and Western Medicine, Nanjing University of Chinese Medicine, Nanjing 210008, China; 3Jiangsu Key Laboratory of Molecular Medicine, Medical School of Nanjing University, Nanjing 210008, China; 4Jiangsu Province Stroke Center for Diagnosis and Therapy, Nanjing 210008, China; 5Nanjing Neuropsychiatry Clinic Medical Center, Nanjing 210008, China; 6Medical School of Nanjing University, Nanjing University, Nanjing 210093, China; 7Drum Tower Hospital Clinical College of Nanjing Medical University, Nanjing 211166, China

**Keywords:** amyotrophic lateral sclerosis, brain network, SC–FC coupling, network-based statistic

## Abstract

Amyotrophic lateral sclerosis (ALS) is a neurodegenerative disorder that progressively affects bulbar and limb function. Despite increasing recognition of the disease as a multinetwork disorder characterized by aberrant structural and functional connectivity, its integrity agreement and its predictive value for disease diagnosis remain to be fully elucidated. In this study, we recruited 37 ALS patients and 25 healthy controls (HCs). High-resolution 3D T1-weighted imaging and resting-state functional magnetic resonance imaging were, respectively, applied to construct multimodal connectomes. Following strict neuroimaging selection criteria, 18 ALS and 25 HC patients were included. Network-based statistic (NBS) and the coupling of grey matter structural–functional connectivity (SC–FC coupling) were performed. Finally, the support vector machine (SVM) method was used to distinguish the ALS patients from HCs. Results showed that, compared with HCs, ALS individuals exhibited a significantly increased functional network, predominantly encompassing the connections between the default mode network (DMN) and the frontoparietal network (FPN). The increased structural connections predominantly involved the inter-regional connections between the limbic network (LN) and the DMN, the salience/ventral attention network (SVAN) and FPN, while the decreased structural connections mainly involved connections between the LN and the subcortical network (SN). We also found increased SC–FC coupling in DMN-related brain regions and decoupling in LN-related brain regions in ALS, which could differentiate ALS from HCs with promising capacity based on SVM. Our findings highlight that DMN and LN may play a vital role in the pathophysiological mechanism of ALS. Additionally, SC–FC coupling could be regarded as a promising neuroimaging biomarker for ALS and shows important clinical potential for early recognition of ALS individuals.

## 1. Introduction

Amyotrophic lateral sclerosis (ALS) is an idiopathic, progressive neurodegenerative disorder that mainly impairs human multimotor systems [1]. Because it begins insidiously with focal weakness but progressively spreads to involve most muscles, early diagnosis is difficult, and the mechanisms underlying the genesis and progression of ALS are poorly understood [1].

With the development and application of multimodal neuroimaging, magnetic resonance imaging (MRI) has enhanced our understanding of disease progression and provided candidate biomarkers for early diagnosis in ALS-related studies [2]. Grey matter-based research has revealed that ALS patients showed focal or widespread atrophy in the motor cortex (e.g., precentral gyrus) and frontal and temporal regions [3,4,5]. White matter microstructural impairments of the corticospinal tract and corpus callosum were also documented in ALS based on diffusion tensor imaging (DTI) [6,7]. Functional connectivity changes in the frontoparietal network (FPN), the subcortical network (SN), and especially the default mode network (DMN) can be used to evaluate cognitive performance and motor function and predict the progression of the disease in ALS patients [8,9,10,11]. Compared to unimodal neuroimaging, multimodal neuroimaging provides more information on the occurrence and progression of ALS. A multimodal imaging study found that by combining white matter microstructural properties, grey matter volume and functional connectivity (FC) helped to discriminate ALS patients from healthy controls (HCs) [12].

Connectomics is a promising method of detecting brain structural and functional connections by modeling multimodal networks [13]. Human brain function is constrained by anatomical interconnections, and the coupling of grey matter structural connectivity (SC) and FC (i.e., SC–FC coupling) can be used to predict FC from its corresponding SC [14]. SC–FC coupling strength can be measured by the correlation between the two networks and provides more sensitive and specific measures of individual functioning than any single modality [14,15], which may lead to the development of better biomarkers. Recent studies have found aberrant SC–FC coupling in patients with attention-deficit/hyperactivity disorder [16], schizophrenia [17,18], bipolar disorder [19], Parkinson’s disease [20], multiple sclerosis [21], and Alzheimer’s disease [22]. Further research found that SC–FC coupling combined with machine learning methods showed better sensitivity and specificity in the diagnosis of diseases, such as major depressive disorder [23,24]. However, few studies have focused on altered grey matter SC–FC coupling and its value for the early diagnosis of ALS.

In the current study, we first applied connectomics techniques to functional and structural imaging data to explore aberrant brain SC and FC in ALS. Second, we combined the support vector machine (SVM) with SC–FC coupling to distinguish ALS patients from HCs and aimed to reveal the underlying neuroimaging mechanisms of ALS.

## 2. Materials and Methods

### 2.1. Participants and Clinical Evaluation

The experimental procedures performed for this research were approved by the ethics committee of involved hospital. All persons gave their informed consent prior to their inclusion in the study. Thirty-seven ALS individuals recruited in 2021 fitted the diagnosis of sporadic possible or probable ALS on the basis of the revised El Escorial criteria [25]. All individuals were assessed by “the revised ALS Functional Rating Scale” (ALSFRS-R) [26]. Clinical variables of “disease duration” and “rate of disease progression” were obtained to reflect disease severity. The “disease duration” measures the time from the symptom onset to the neuroimaging assessment. The “rate of disease progression” is computed by linearly averaged decrease of ALSFRS-R scores per month from the symptom onset [(48-ALSFRS-R)/(disease duration)]. The exclusion criteria for ALS individuals included family history of neurodegenerative diseases (including a family history of ALS), frontotemporal dementia [27], cognitive impairment, major psychiatric and other neurological disorders. Twenty-five HCs with no history of neuro-psychiatric illness were included. Diagram flow of the sample composition was described in Figure 1. All individual subjects provided written informed consent.

### 2.2. MRI Acquisition

All participants underwent multimodal neuroimaging scanning on a 3.0T Philips MRI (Philips Medical Systems, Eindhoven, The Netherlands). Resting-state functional MRI (rs-fMRI) data were acquired by the gradient-recalled echo planar imaging sequence [repetition time (TR) = 2000 ms, flip angle (FA) = 90°, echo time (TE) = 30 ms, number of slices = 35, thickness = 4.0 mm, field of view (FOV) = 240 × 240 mm^2^, acquisition matrix = 64 × 64]. For structural MRI data, high-resolution 3D T1-weighted images were performed in a turbo gradient echo sequence [TR = 9.8 ms, FA = 8°, TE = 4.6 ms, number of slices = 192, thickness = 1.0 mm, FOV = 250 × 250 mm^2^, acquisition matrix = 256 × 256].

### 2.3. MRI Data Preprocessing and Multimodal Connectome Construction

#### 2.3.1. Functional Connectome

The rs-fMRI data in this study were preprocessed by statistical parametric mapping (SPM12, http://www.fil.ion.ucl.ac.uk/spm/software/spm12/ (assessed on 9 September 2022)) and Data Processing & Analysis for Brain Imaging (DPABI V4.1, http://rfmri.org/dpabi/ (assessed on 9 September 2022)). The major preprocessing steps included slice-timing and head motion correction, normalization to the Montreal Neurological Institute space, multiple linear regression analysis and band-pass filtering (0.01–0.1 Hz). In addition, subjects who exhibited an angular rotation >2°, a displacement >2 mm in any direction, or mean Jenkinson frame-wise displacement (FD) greater than 0.2 mm were excluded from this study [28]. The entire brain was divided into 246 cortical and subcortical regions based on the Human Brainnetome Atlas (https://atlas.brainnetome.org/ (assessed on 9 September 2022), the detailed brain segmentation in Appendix A). Pearson correlation between the regional mean time series of each pair of brain regions was conducted. Only those functional connectivity whose corresponding *p* values lower than 0.05 (false discovery rate-corrected) were preserved [29]. Finally, we constructed a 246 × 246 symmetric correlation matrix to represent the individual functional network (Figure 2).

#### 2.3.2. Structural Connectome

Before constructing the grey matter structural network, the 3D T1 imaging were preprocessed by using the Computational Anatomy Toolbox (CAT12, http://www.neuro.uni-jena.de/cat/ (assessed on 9 September 2022)) as implemented in the SPM12. The main preprocessing steps included correction for bias-field inhomogeneities, tissue segmentation into white matter, grey matter and cerebrospinal fluid, and spatial normalization with the DARTEL algorithm. Then, the grey matter volume maps were smoothed spatially (Gaussian kernel with 6 mm full width at half maximum). To construct grey matter structural connectome, the Jensen–Shannon distance-based similarity (JSS) was applied to define structural connectivity between each pair of brain regions [30]. For each subject, we first extracted the grey matter volume values of all voxels in each brain region. Second, the probability density function of these grey matter volume values was estimated by the kernel density. Third, the probability distribution function (PDF) was computed according to the above probability density function. Afterwards, the JSS between each pair of brain regions was computed by their PDF and ranged from 0 to 1. Finally, a 246 × 246 symmetric similarity matrix was obtained to represent the grey matter structural connectome of each participant (Figure 2).

### 2.4. Network-Based Statistic Analysis

To identify the specific subnetwork that showed a significant group difference between ALS and HC patients, we applied a network-based statistic (NBS) approach [31]. Based on the step of permutation, the NBS could correct for the family-wise error due to the large number of connections compared. Briefly, a primary threshold (*p* < 0.001, uncorrected) was applied to the two-sample t test computed for each edge to define a set of suprathreshold connections, among which any connected subnetworks and their sizes were then determined. A corrected *p* value was computed for each connected component using the null distribution of the maximal component size, which was empirically derived by using a nonparametric permutation approach (5000 permutations). The 246 brain regions from the Human Brainnetome Atlas were mapped into the 7 functional network atlases proposed by Yeo et al. and the subcortical network [32]. The eight intrinsic brain networks included the SN, limbic network (LN), visual network (VN), somatomotor network (SMN), FPN, DMN, dorsal attention network (DAN), and salience/ventral attention network (SVAN). Finally, we counted the number of connections with significant statistical differences within and between subnetworks.

### 2.5. Calculation of SC–FC Coupling

A correlation analysis was conducted between the strengths of the SC and FC for the whole-brain SC–FC coupling. All nonzero values of the SC matrix were selected and rescaled to a Gaussian distribution. The new SC vector was correlated with its functional counterparts derived from the FC matrix. Then, a single whole-brain SC–FC coupling metric for each participant was obtained, indicating the SC–FC coherence across the whole-brain regions. Regional-node SC–FC coupling was performed by computing the correlation coefficient between a row of the SC matrix with its corresponding row of the FC matrix (excluding the self-connection). This resulted in a vector of 246 values that represented the regional-node SC–FC coupling for each of the 246 regions in each participant. In addition, we separately calculated the global-network SC–FC coupling, within-network SC–FC coupling, and between-network SC–FC coupling from the perspective of the brain network. The global-network SC–FC coupling for each subnetwork was calculated by correlating the SC matrix of each subnetwork with the corresponding FC matrix. The within-network SC–FC coupling for each brain region was the correlation of the FC and SC between that region and other regions in the same subnetwork. The between-network SC–FC coupling for each brain region was performed between that region and other regions belonging to other subnetworks. The pattern diagrams of these SC–FC coupling measures are shown in Figure 2.

### 2.6. Machine Learning

The SVM with linear kernel was performed by using a toolkit (LIBSVM, http://www.csie.ntu.edu.tw/cjlin/libsvm/ (assessed on 9 September 2022)). The SVM analysis involves three steps: feature selection, classifier training, and prediction. In our research, statistically significant features, such as regional-node SC–FC coupling, global-network SC–FC coupling, within-network SC–FC coupling, and between-network SC–FC coupling, were first selected for SVM to form a high dimensional space (Figure 2). Second, SVM conducts the classifier training to construct a hyperplane that optimally separates the classes. Last, the classifier is used to predict the class label when a new sample is added into the classifier. Due to the limited number of samples, we used the leave-one-out cross-validation (LOOCV) scheme to assess the performance of the classifier. The performance of a classifier can also be quantified using sensitivity, specificity, and accuracy. Note that the specificity represents the proportion of HC subjects correctly predicted, while the sensitivity represents the proportion of ALS subjects correctly predicted. To further evaluate the statistical significance of the observed classification accuracy, we conducted permutation testing. During each permutation test, we randomly permuted the labels of participants prior to SVM analysis, and LOOCV was then performed on the permuted datasets. This procedure was repeated 5000 times to determine whether the classification accuracy occurs by chance. To achieve stable classification performance, we also applied the linear discriminant analysis and the bagged trees, which were performed in MATLAB 2017b. The detailed information is described in the Appendix A.

### 2.7. Statistical Analysis

Statistical analyses of demographic characteristics were performed using Statistical Package for Social Sciences software (SPSS V22, IBM Corp., Armonk, NY, USA). We used a Chi-square (χ^2^) test to test group differences in gender and applied a two-sample t test for age and education. To compare these SC–FC coupling measures between the HC and ALS groups, a nonparametric permutation test was used and repeated 5000 times. *p* < 0.05 was considered as a significant level (uncorrected). The permutation procedure was conducted based on the permutation code embedded in the Graph Theoretical Network Analysis Toolbox (GRETNA, http://www.nitrc.org/projects/gretna/ (accessed on 9 September 2022)).

## 3. Results

### 3.1. Demographic and Clinical Characteristics

In the ALS group, nineteen ALS individuals were excluded from analysis because of lacking multimodal MRI data (n = 13), excessive head movement (n = 4), and cerebral vascular damage (n = 2) (Figure 1). Finally, 18 ALS and 25 HC patients were involved in the imaging analyses. Demographic and clinical data are provided in Table 1. There were no significant differences in the age, gender, and Montreal cognitive assessment (MoCA) between the ALS and HC groups.

### 3.2. Altered Functional Network Connectivity in ALS

Compared to the HC group, the NBS result identified a single subnetwork containing 39 nodes with 49 significantly increased connections in the ALS group (*p* = 0.019, corrected, Figure 3A). This subnetwork predominantly encompassed the connections between the DMN and the FPN (edges: 11/49, 22.45%), and the DMN and the DAN (edges: 9/11, 18.37%).

### 3.3. Altered Structural Network Connectivity in ALS

Non-parametric NBS analysis showed two subnetworks with significant differences in SC matrices between the ALS and HC groups. Compared to HCs, ALS patients showed significantly increased connections within the first subnetwork (*p* = 0.009, corrected, the upper row of Figure 3B). This subnetwork consisted of 129 nodes and 202 edges, predominantly involved the inter-regional connections between the LN and the DMN (edges: 36/202, 17.82%), the LN and the SVAN (edges: 24/202, 11.88%), and the LN and the FPN (edges: 22/202, 10.89%). Compared to the HC group, the second subnetwork was composed of 73 nodes and 168 edges, which was significantly decreased in the ALS group (*p* = 0.009, corrected, the bottom row of Figure 3B). This subnetwork mainly involved the connections between the LN and the SN (edges: 60/168, 35.71%).

### 3.4. SC–FC Coupling Measures and Their Clinical Prediction

There was no significant difference in the whole-brain SC–FC coupling between the ALS and HC groups (*p* = 0.29, 5000 permutations). We separately calculated the global-network SC–FC coupling from the perspective of brain network. The ALS group exhibited significantly increased global-network SC–FC coupling strength in the SN (*p* = 0.024, 5000 permutations), SMN (*p* = 0.025, 5000 permutations), and DMN (*p* = 0.009, 5000 permutations) compared to the HC group, as shown in Figure 4A. No significant difference in global-network SC–FC coupling was observed in other subnetworks (e.g., LN, VN, FPN, DAN, and SVAN). Based on the features of global-network SC–FC coupling, the SVM model could distinguish the ALS patients from HCs with an accuracy of 72.09%, a sensitivity of 88.00%, and a specificity of 61.11% (*p* = 0.008, 5000 permutations, Figure 4A).

From the perspective of node-level SC–FC coupling, ALS patients exhibited significantly different regional-node SC–FC coupling in 23 brain regions compared to the HC group (*p* < 0.05, 5000 permutations, the upper row of Figure 4B). Among them, the regional-node SC–FC coupling of LN-related brain regions (e.g., hippocampus and parahippocampal gyrus) was significantly reduced, and the regional-node SC–FC coupling of DMN-related brain regions (e.g., prefrontal lobe and anterior cingulate gyrus) was significantly enhanced in the ALS group. By applying node-level SC–FC coupling as features, the SVM model showed classification performance with an accuracy of 81.40%, a sensitivity of 80.00%, and a specificity of 94.44% (*p* = 0.001, 5000 permutations, the upper row of Figure 4B), and the node-level SC–FC coupling in the parahippocampal gyrus represented the best predictive value.

ALS patients showed significantly different within-network SC–FC coupling in 17 brain regions compared with HCs (*p* < 0.05, 5000 permutations, the middle row of Figure 4B). A distribution pattern similar to that of regional-node SC–FC coupling was observed. The within-network SC–FC coupling of LN-related brain regions was significantly reduced, while the within-network SC–FC coupling of DMN-related brain regions was significantly enhanced in the ALS group. In addition, the within-network SC–FC coupling could be applied to distinguish ALS patients from HCs with an accuracy of 76.74%, a sensitivity of 72.00%, and a specificity of 94.44% (*p* = 0.005, 5000 permutations, the middle row of Figure 4B). From the perspective of between-network SC–FC coupling, ALS patients exhibited significantly different regional-node SC–FC coupling in 19 brain regions compared with HCs (*p* < 0.05, 5000 permutations, the bottom row of Figure 4B). The distribution pattern of these 19 brain regions was similar to the regional-node and within-network SC–FC coupling mentioned above. By using the between-network SC–FC coupling as features, the SVM analysis showed a classification performance with an accuracy of 76.74%, a sensitivity of 80.00%, and a specificity of 72.22% (*p* = 0.005, 5000 permutations, the bottom row of Figure 4B).

Due to the effects of overfitting and small sample size on our results, we also performed a linear discriminant analysis and the bagged trees. In general, these two models preliminarily showed similar classification performance compared with the SVM, as described it in the Appendix A.

## 4. Discussion

In this study, using multimodal neuroimaging techniques, we constructed functional and structural networks and examined altered brain networks in patients with ALS. In addition, we employed a hierarchical analysis approach (from global to regional scales) to investigate the abnormal pattern of SC–FC coupling in ALS and combined these data with the SVM approach to distinguish ALS patients from HCs. Our findings revealed that ALS patients showed significantly increased functional connectivity of interconnected edges, primarily related to the DMN. In the structural network-based analyses, the enhanced grey matter structural connections predominantly involved the interregional connections between the LN and other cortical networks, while the weakened structural connections mainly involved connections between the LN and the SN. We also found stronger SC–FC coupling in DMN-related brain regions and decoupling in LN-related brain regions in ALS patients based on the SVM approach, which is promising for distinguishing ALS patients from HCs.

The NBS results indicated that a single connected subnetwork exhibited increased connections in the ALS group compared with the HC group. The subnetwork predominantly encompassed the connections between the DMN and other cortical networks (e.g., the FPN and DAN). Altered resting-state functional network connectivity, especially in the DMN, in ALS has been extensively reported in previous studies [9,10,11,33,34,35,36]. However, fMRI studies have reported inconsistent findings. A few studies were consistent with our findings and reported increased functional connectivity in the DMN (e.g., parahippocampal, frontoparietal, prefrontal, cingulate, and thalamic regions) in ALS patients compared to controls. Furthermore, the strength of functional connectivity in the DMN of ALS patients exhibited more significant correlations with cognitive function, disability, and disease progression, which might indicate physiological compensation for impaired structural integrity [11,33,35]. However, other research has described hypoconnectivity in DMN-related regions, including the cingulate cortex, inferior parietal cortex, superior mid-frontal gyrus, and frontotemporal gyrus, which have been linked previously to executive functions and global cognition [9,10,36,37]. Notably, the above-mentioned studies investigated functional abnormalities at a regional level. We employed a network-based neuroimaging method to determine how internetwork and intranetwork connectivity was altered in ALS and to expand upon previous neuroimaging findings. It is well established that disease heterogeneity, severity, and rate of progression differ significantly among studies, which may have also caused the differences in the results.

In our study, we demonstrated that the stronger grey matter structural connectivity predominantly involved the interregional connections between the LN and other cortical networks, while the weakened grey matter structural connectivity mainly involved the connections between the LN and the SN. It is worth noting that ALS patients showed more widespread white matter structural alterations than functional alterations compared with HCs, especially in terms of damage [36,38,39]. These findings indicate that structural alterations may occur earlier than functional abnormalities. As a major relay station that modulates input from many cortical areas, the thalamus (an important region of the SN) is involved in the pathophysiology of ALS [40]. Abnormalities in the thalamus on diffusion MRI, MR spectroscopy, and fMRI have been reported in many studies; these abnormalities are mostly consistent with our findings [40,41]. According to the network-based hypothesis, pathological alterations could physically spread along neuroanatomical connections in the brain. Thus, it is reasonable to assume that altered functional connectivity may lead to damage to the brain structural network. Previous studies investigated the correlation between changes in FC and white matter SC of the brain network. We speculated that the coupling analysis of SC and FC, both originating from grey matter, would be more consistent. To our knowledge, no studies have focused on altered grey matter structural connectivity in ALS. We hypothesize that the compensation of changes in the DMN-related functional network and the LN-cortical-related brain structural network may follow the structural impairment of LN–SN connectivity in the progression of ALS.

Combining the analyses of grey matter and functional networks, we explored ALS pathophysiology from the perspective of SC–FC coupling strength on global and regional scales. Our main findings were that ALS patients exhibited stronger SC–FC coupling in DMN-related brain regions and weaker SC–FC coupling in LN-related brain regions. Estimating the correspondence between SC and FC matrices at different spatiotemporal scales, known as SC–FC coupling, has been proposed for the quantification of this correspondence. Stronger SC–FC coupling suggests a more direct association between FC and SC. In contrast, decoupling (i.e., weaker coupling) may suggest a loss of coherence. The opposite trends of SC–FC coupling between the DMN and the LN indicate the nonsynchronous disruption of structural and functional connectivity. This is consistent with previous studies that showed widespread structural (but not functional) damage in ALS patients compared with HCs [36,39]. Incorporating the significant differences in SC–FC coupling as features in the SVM model resulted in classification performance with similar or even higher accuracy (72.09–81.40%), sensitivity (72.00–88.00%), and specificity (61.11–94.44%) for distinguishing ALS patients from HCs compared to that obtained in previous studies using other neuroimaging features [42,43,44,45]. Our findings show that SC–FC coupling may be a valuable neuroimaging marker for ALS.

Several limitations of the present investigation should be considered. First, ALS sample size was relatively small and the attrition rate, due to low MRI image quality, was high (i.e., about half of the studied population), hindering the monitoring of fast progressors and the identification of multi-modal mechanisms of ALS. Furthermore, multiple comparison corrections were not performed when comparing SC–FC-coupled between-group differences. These findings should be replicated in future multicenter longitudinal datasets. Second, other MRI modalities (e.g., DTI) can also help to identify ALS at the early stage [43]; thus, multimodal neuroimaging combinations should be considered in future research. Third, we did not determine the *C9orf72* repeat expansion status of ALS patients in this study. Recognition of the ALS patients who carry the *C9orf72* repeat expansion is important in the context of appropriate disease management and stratification in clinical trials. In future research, we will perform genetic analysis and explore the *C9orf72* repeat expansion status in ALS patients. Fourth, ALS is gradually characterized as a multisystem disorder, which may why linear models exhibit poor performance. We hope to improve the model performance by applying other methods (e.g., a graph neural network).

## 5. Conclusions

This study demonstrated that ALS individuals exhibited a characteristic pattern of grey matter functional and structural connectivity changes in the DMN and the LN. Stronger DMN-related functional connectivity and the coexistence of stronger and weaker LN-related structural connectivity were observed in ALS patients. Additionally, we found stronger SC–FC coupling in DMN-related brain regions and decoupling in LN-related brain regions in ALS patients based on the SVM approach, which is promising for distinguishing ALS patients from HCs. Our findings suggest that SC–FC coupling strength is a valuable biological feature for diagnosing ALS in clinical practice.

## Figures and Tables

**Figure 1 brainsci-13-00803-f001:**
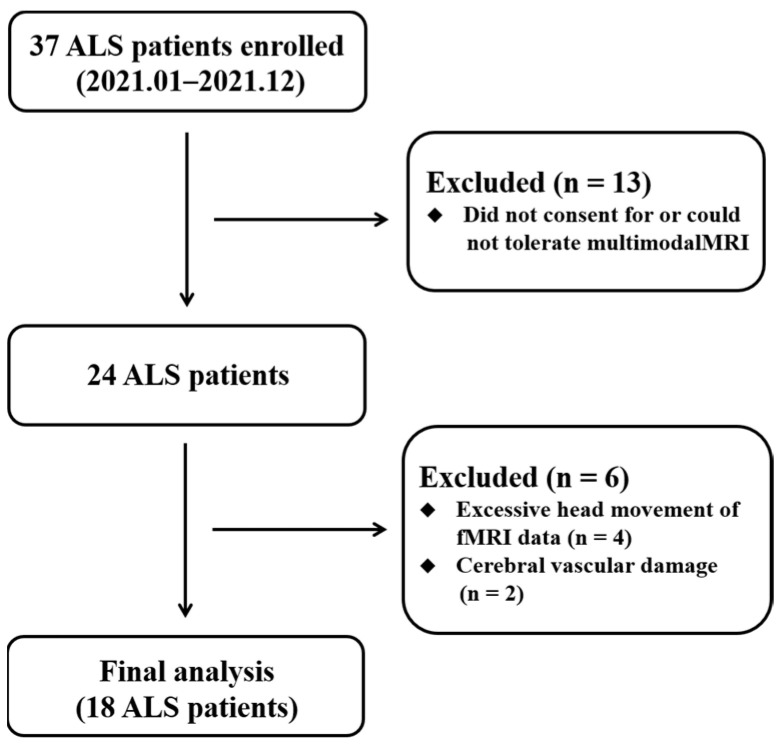
Diagram flow of the ALS recruitment. Abbreviations: ALS, amyotrophic lateral sclerosis.

**Figure 2 brainsci-13-00803-f002:**
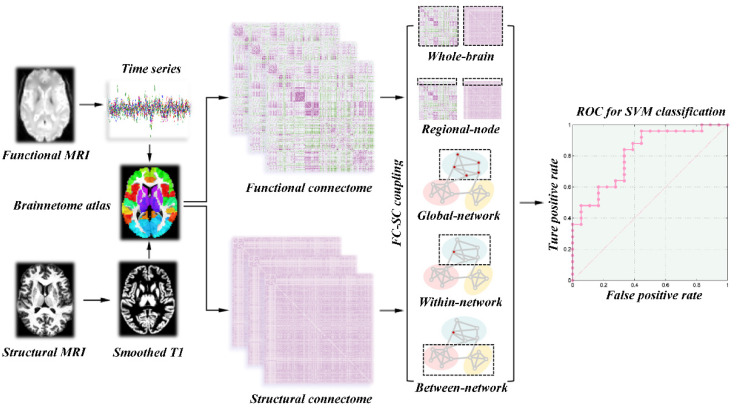
Workflow for quantifying SC–FC coupling and SVM analysis. Abbreviations: SC, structural connectivity; FC, functional connectivity; SVM, support vector machine.

**Figure 3 brainsci-13-00803-f003:**
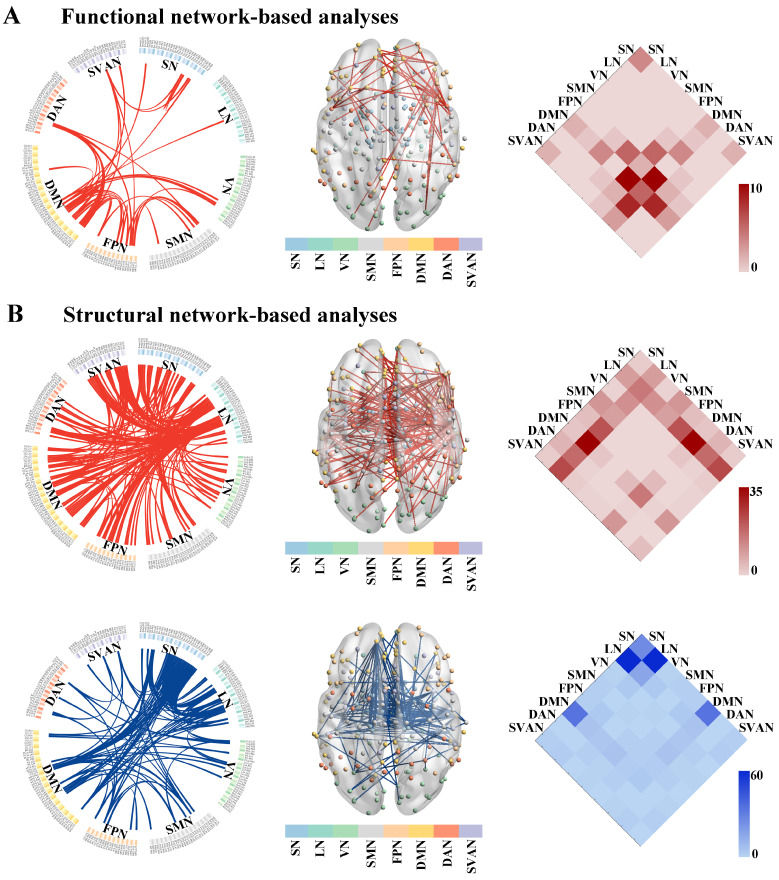
The altered connected subnetwork in patients with ALS based on the NBS analysis. (**A**) Altered functional network connectivity in ALS. A single connected subnetwork containing 39 nodes with 49 significantly increased connections in patients with ALS compared with the HC group (*p* = 0.019, corrected). (**B**) Altered grey matter structural network connectivity in ALS. ALS patients showed significantly increased connections within the first subnetwork consisting of 129 nodes and 202 edges (*p* = 0.009, corrected, the upper row of (**B**)). The second subnetwork was composed of 73 nodes and 168 edges, which was significantly decreased in ALS compared with the HC group (*p* = 0.009, corrected, the bottom row of (**B**)). Abbreviations: ALS, amyotrophic lateral sclerosis; HC, healthy control; NBS, network-based statistic; SN, subcortical network; LN, limbic network; VN, visual network; SMN, somatomotor network; FPN, frontoparietal network; DMN, default mode network; DAN, dorsal attention network; SVAN, salience/ventral attention network.

**Figure 4 brainsci-13-00803-f004:**
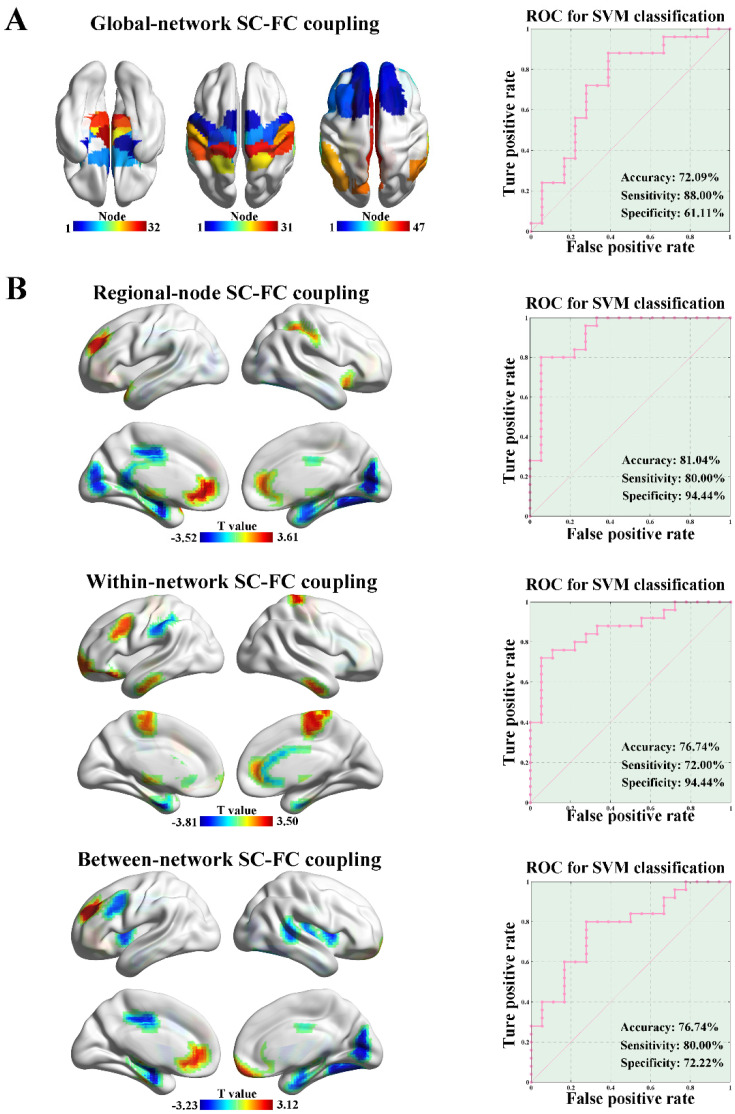
SC–FC coupling and clinical predictive analysis. (**A**) The ALS group showed significantly increased global-network SC–FC coupling than the HC group in the SN (*p* = 0.024, 5000 permutations), SMN (*p* = 0.025, 5000 permutations) and DMN (*p* = 0.009, 5000 permutations). Based on the features of global-network SC–FC coupling, SVM analysis was able to classify the ALS subjects from HCs with an accuracy of 72.09%, a sensitivity of 88.00%, and a specificity of 61.11% (*p* = 0.008, 5000 permutations). (**B**) ALS patients exhibited significantly different regional-node SC–FC coupling in 23 brain regions compared with HCs (*p* < 0.05, 5000 permutations, the upper row of (**B**)). By applying the node-level SC–FC coupling as features, the SVM model showed the classification performance with an accuracy of 81.40%, a sensitivity of 80.00%, and a specificity of 94.44% (*p* = 0.001, 5000 permutations, the upper row of (**B**)). ALS patients showed significantly different within-network SC–FC coupling in 17 brain regions compared with HCs (*p* < 0.05, 5000 permutations, the middle row of (**B**)). The within-network SC–FC coupling could be applied to distinguish ALS patients from HCs with an accuracy of 76.74%, a sensitivity of 72.00%, and a specificity of 94.44% (*p* = 0.005, 5000 permutations, the middle row of (**B**)). From the perspective of between-network SC–FC coupling, ALS patients exhibited significantly different regional-node SC–FC coupling in 19 brain regions compared with HCs (*p* < 0.05, 5000 permutations, the bottom row of (**B**)). By using the between-network SC–FC coupling as features, the SVM analysis showed the classification performance with an accuracy of 76.74%, a sensitivity of 80.00%, and a specificity of 72.22% (*p* = 0.005, 5000 permutations, the bottom row of (**B**)).

**Table 1 brainsci-13-00803-t001:** Demographic and clinical data in ALS patients and HCs.

Items	HC (n = 25)	ALS (n = 18)	Statistical Value	*p* Value
Age (years)	53.40 ± 2.89	54.11 ± 12.09	−0.24	0.78 ^b^
Gender (male/female)	17/8	9/9	1.42	0.23 ^a^
MoCA	27.88 ± 1.13	27.33 ± 0.84	1.73	0.09 ^b^
Disease duration at baseline (months)	--	9.97 ± 8.45	--	--
Site of onset (spinal/bulbar)	--	16/2	--	--
ALSFRS-R at baseline (0–48)	--	41.89 ± 4.00	--	--
ALSFRS-R progression rate [(48-ALSFRS-R score)/disease duration]	--	1.06 ± 0.97	--	--
Diagnostic category (Definite/probable/probable lab-supported/possible)	--	14/3/1/0	--	--

Values are presented as the mean ± standard deviation. ^a^
*p* value was obtained by the χ^2^ test, ^b^
*p* value was obtained by two-sample *t* tests. Abbreviations: ALS, amyotrophic lateral sclerosis; HC, healthy control; MoCA, Montreal cognitive assessment; ALSFRS-R, the revised ALS functional rating scale.

## Data Availability

If anyone is interested in extrapolating their data for further validation, the MRI images can be made available to the scientific community from the corresponding author.

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
