# Peer review of "Aberrant Multimodal Connectivity Pattern Involved in Default Mode Network and Limbic Network in Amyotrophic Lateral Sclerosis"

_brainsci, 2023, doi:10.3390/brainsci13050803_

Round 1

Reviewer 1 Report

Chen et al. studied the similarity between brain regions (which was called structural connectivity) and resting state functional connectivity in 25 controls and 18 ALS patients. They used a support vector machine to separate cases and controls. I have some comments on this study:

1.     Structural connectivity is generally used for the white matter connectome (as measured by diffusion tensor imaging (DTI) for example). Because the authors did not perform DTI but determined “structural connectivity” based on the similarity of brain regions it is more precise and better to use a term such as structural similarity connectivity or grey matter structural connectivity, or some other term that makes clear that the authors did not perform DTI.

2.     The amount of information in Table 1 is quite limited. Some more information is needed. For example, information about cognitive and behavioural performance, site of onset, classification according to the El Escorial criteria (which information is available based on the text, but not given in Table 1), survival, etc.

3.     Although the C9orf72 repeat expansion is known to be quite rare in the Chinese population, its impact on imaging is big and this should therefore be reported, or if not possible, be mentioned as a limitation in the discussion.

4.     Sample size is small, as mentioned by the authors in the final paragraph of the discussion, but the authors only drop this observation without commenting on its implications. I would encourage the authors to provide some implications of the small sample size for the interpretation of this study.

5.     Support vector machines are well-known for their overfitting. This applies even more to small sample sizes and large numbers of predictors. In this setting, it would be more robust to use (for example) some form of penalized regression.

6.     I missed a paragraph with a direct comparison with other multimodal neuroimaging studies in ALS that assessed structural and functional connectivity and the coupling between both. What are the similarities and differences? And why?

7.     In the last sentence, ALS is misspelt (as well as in other sentences in this manuscript).

Reviewer 2 Report

The paper tries to identify differences between ALS (n=18) and healthy controls (n=25) using multi-modal connectomes (structural and functional brain connectivity). Changes in the functional connectivity (FC) and structural connectivity (SC) were identified using the network-based statistics (NBS) approach and subsequently a coupling of the structural-functional (SC) connectivity was computed. Different metrics for the SC-FC coupling were used with a linear SVM classifier to differentiate between the ALS and control groups. Results show a positive ability of the multi-modal connectome to separate the ALS and controls, while highlighting the differences in the SC, FC and SC-FC coupled networks of the two groups.   Strengths - Overall, I think that the work tried to use simple metrics with linear classifiers which has helped with the model interpretability.    Weaknesses -
  1. The study lacks an independent validation cohort. It will be important to see how the results are on an independent dataset using just the selected features from this study. 
  2. Will be important to estimate the variance of the ROC-AuC and the associated metrics (accuracy, sensitivity etc.). A k-fold cross validation would help to look at the variance with the ROC curves (The leave one out cross-validation which has been used does not permit looking at the variance of the ROC curves since the test set has only one sample in each iteration).
  3. Was regularization used with the SVM? Given that the input is high dimensional and there aren't too many samples, regularizing the SVM model will be important.
  4. In the feature selection step prior to applying SVM, are the same features re-selected if a different feature selection step is used such as LASSO which provides regularization and feature selection implicitly? An analysis into the reliability of the selected features, and the brain regions represented by them will be important.
  5. Doesn't seem very clear how the following coupling metrics were computed - global-network, within-network and between-network coupling metrics. Were the functional entries taken from the overall functional connectivity matrix or the overlap with functional subnetwork matrix?
  6. Instead of using the curated SC-FC coupling metrics (global/within/between network), have the authors considered using a graph-neural network approach which will compute the features in an end-to-end fashion? Does that outperform the linear SVM in this case?
  7. Many works have suggested ALS is not well modeled by linear methods. A brief discussion should be added to the paper on the limitations.

Reviewer 3 Report

The manuscript by Chen et al. shows fMRI-based structural and functional connectivity changes in ALS patients in comparison to healthy controls.

Overall, the authors provide interesting data for the field of motor neuron disease.

Some aspects of the manuscript have to be reconsidered:

1) The small number of ALS patients somewhat puts the study`s interpretive power into perspective, as the authors themselves note.

2) Half of the ALS cohort could not be included in the study, which seems a bit much

3) Because of the frequent association of ALS and FTLD, it is somewhat unfortunate that these patients were excluded

4) It would be helpful if the findings found were discussed in the overall concept of ALS. Are these primary changes taking place or secondary effects? Wouldn`t it be better to conduct several studies staggered in time on the same subjects?

Round 2

Reviewer 1 Report

The authors made adequate adjustments. Two points I mentioned in my previous report are not fully solved:

5. The authors repeat the methods used. My point is, however, that these methods (support vector machines) are sensitive to overfitting and that other methods (such as penalized regression) are more appropriate in this setting, especially with the relatively small sample size. The number of permutations performed does not change this view. As far as I could see, this point was also raised by another reviewer and therefore needs extra attention from the authors. If the current form is not changed, I would suggest adding some additional sentences in the discussion to underscore the risk of overfitting the support vector machines, especially in this relatively small sample.

6. I thank the authors for their extensive reply, however, the sentence "(...) no studies have focused on altered grey matter SC and SC-FC coupling in ALS" is incorrect. For example, Schmidt et al. (Correlation between structural and functional connectivity impairment in amyotrophic lateral sclerosis) reported in 2014 on this subject.

Reviewer 2 Report

The authors have made some clarifications in the limitations but have done little to address the technical deficiencies pointed out by both reviewers.  If the authors are insisting on this type of study design, there needs to be another model baseline to insure the results are stable.  The choice of SVM with a one hold out selection and using t-tests for filtering does not meet the usual required technical veracity for such machine learning algorithms in the presence of small samples.  Moreover, the authors' explanations are no sufficient to suggest these choices are appropriate without assessing further model methodology (non-SVM method, different form of regularization and optimization, and a different of feature filtering). Each of these choices singly and in combination could be greatly influencing results in such a small sample set.  A more comprehensive assessment of methodological choices is necessary to insure veracity.  Without this assessment, the machine learning adds little more than standard network statistics.

Reviewer 3 Report

No concerns.

Author Response

Thank you.